# The Protective Role of Butyrate against Obesity and Obesity-Related Diseases

**DOI:** 10.3390/molecules26030682

**Published:** 2021-01-28

**Authors:** Serena Coppola, Carmen Avagliano, Antonio Calignano, Roberto Berni Canani

**Affiliations:** 1Department of Translational Medical Science, University of Naples Federico II, 80131 Naples, Italy; serenacoppola@mail.com; 2ImmunoNutriton Lab at CEINGE Advanced Biotechnologies, University of Naples Federico II, 80131 Naples, Italy; 3Department of Pharmacy, University of Naples Federico II, 80131 Naples, Italy; carmen.avagliano@unina.it (C.A.); calignan@unina.it (A.C.); 4European Laboratory for the Investigation of Food Induced Diseases (ELFID), University of Naples Federico II, 80131 Naples, Italy; 5Task Force on Microbiome Studies, University of Naples Federico II, 80131 Naples, Italy

**Keywords:** short-chain fatty acids, butyric acid, metabolic diseases, gut microbiota

## Abstract

Worldwide obesity is a public health concern that has reached pandemic levels. Obesity is the major predisposing factor to comorbidities, including type 2 diabetes, cardiovascular diseases, dyslipidemia, and non-alcoholic fatty liver disease. The common forms of obesity are multifactorial and derive from a complex interplay of environmental changes and the individual genetic predisposition. Increasing evidence suggest a pivotal role played by alterations of gut microbiota (GM) that could represent the causative link between environmental factors and onset of obesity. The beneficial effects of GM are mainly mediated by the secretion of various metabolites. Short-chain fatty acids (SCFAs) acetate, propionate and butyrate are small organic metabolites produced by fermentation of dietary fibers and resistant starch with vast beneficial effects in energy metabolism, intestinal homeostasis and immune responses regulation. An aberrant production of SCFAs has emerged in obesity and metabolic diseases. Among SCFAs, butyrate emerged because it might have a potential in alleviating obesity and related comorbidities. Here we reviewed the preclinical and clinical data that contribute to explain the role of butyrate in this context, highlighting its crucial contribute in the diet-GM-host health axis.

## 1. Introduction

Obesity is a public health concern worldwide. It has been defined by the World Health Organization as abnormal or excessive fat accumulation that may impair health [1]. This condition has reached pandemic levels, in the last four decades its prevalence has tripled: about 39% of the world’s adult population is actually overweight and 13% (11% of men and 15% of women) is obese [1]. This health issue affects all age groups and countries of all income levels. Globally there are more people who are obese than underweight, this occurs in every area except parts of sub-Saharan Africa and Asia, and actually obesity is linked to more deaths worldwide than underweight [1]. The onset of obesity in childhood is a risk factor for adult obesity and is linked to significant negative physical and psychosocial consequences: last data reported that 38 million children under 5 years are overweight or obese and over 340 million children and adolescents are overweight or obese [1]. So, intervening in children is essential to change their weight trajectory and try to prevent adult obesity and his health sequelae. This largely preventable health threat is a risk factor for other metabolic diseases such as type 2 diabetes, cardiovascular diseases, dyslipidemia, non-alcoholic fatty liver disease, chronic kidney disease, obstructive sleep apnea and hypoventilation syndrome, mood disorders and physical disabilities [2]. A very small proportion of obesity cases result from monogenic alterations (syndromic and non-syndromic), so the common forms of obesity are multifactorial and are most likely due to a complex interplay of environmental changes (obesogenic environment) and the individual genetic predisposition [3]. The main drivers of multifactorial obesity pathogenesis seem to be a long-term of energy discrepancy between too many calories consumed and an increase of sedentary behavior [4]. Behind this, the over-representation of genetic variants that favor overeating and/or low energy expenditure can be sought in the selective pressure that has most likely contributed to the evolution of a human genotype able to preserve the few foodstuffs available to survive in a long period of famine and then exposed to an increasingly obesogenic environment [5]. One of astonishing findings is the role played by human gut microbes (commonly defined as gut microbiota, GM) that could represent the causative link between environmental factors and the onset of obesity, since a growing body of evidence suggests that the set of microorganisms that live within the digestive tract, making up the GM, play a role in energy regulation and substrate metabolism [6]. Germ-free mice offer one of key finding for determining the role of the GM in energy balance: these experimental animal models are protected from diet-induced obesity when compared with conventional (colonized) counterparts, despite consuming more calories [7]. Furthermore, fecal transplant from obese humans to germ-free mice elicit more weight gain than mice that receive microbes from normal weight humans [8]. Dysbiosis, a condition of aberrant GM, induces metabolic, inflammatory and immune disturbances both locally and, consequent upon impaired gut barrier function, also systemically; so, the gut microecology could thus fill the gap between energy intake and expenditure by processing nutrients and regulating their access to and storage in the body, through the secretion control of hormones and mediators of energy homeostasis [9]. Conversely, being highly sensitive to environmental impacts, particularly to diet, the development of the GM may prove the target of choice in efforts to reduce the risk of obesity. The number of metagenomic data generated on obese subjects can lead to the erroneous claim that a bacterium is causally linked with the protection or the onset of this disease, in fact, the necessity for analyzing not only the presence of certain gut microorganisms but also their activity (including the metabolome) should be considered [10]. Among the factors which impact the GM, dietary compounds deeply affect the growth and metabolism of gut bacteria, since fermentation of nutrients is one core function of the intestinal microbes [11]. Within fermentation products an array of small organic metabolites is short-chain fatty acids (SCFAs) acetate, propionate and butyrate [12]. SCFAs are the primary end products of degradation of dietary fibers, the fraction of not digested carbohydrates (CHO) by endogenous enzymes in the small intestine; in the context of indigestible CHO some fibers are not used by human gut microbes, such as cellulose, whereas some CHO are fermented by gut bacteria (such as resistant starch (RS)) but fall outside fibers definition [13]. The “Westernization” of lifestyles might lead to an increase in obesity levels, so GM investigation made progress in understanding the role of resident bacteria in relation to Western metabolic diseases [14]. Microbiota-accessible carbohydrates (MACs) are CHO available to gut microbes’ degradation that are notably reduced in the Western-style diet, which is high in fat and simple CHO and low in fibers [15]; furthermore, evidence show that microbiota transplantation from mice fed a Western diet to germ-free mice transfers the obese phenotype [16]. Another consequence of MACs restriction, typical of Western diet, is a reduction in SCFAs production with the consequent loss of SCFAs benefits, potent regulatory molecules with vast physiological effects such as energy metabolism, intestinal homeostasis and immune responses regulation [13]. However, it is clear that three SCFAs differ substantially in their potential effects upon host physiology. Firstly, they differ in their fate and tissue distribution, with butyrate as the main energy source for colonocytes, propionate which contribute to liver gluconeogenesis and acetate that achieves the highest systemic concentrations in blood; secondly, they differ in the interaction with host proteins (e.g., inhibition of histone deacetylases by butyrate and propionate) and receptors [17]. Among SCFAs, butyrate emerged because it might have a potential in alleviating obesity and related metabolic complications typical of westernized countries [18]. For demonstration, a lower abundance of butyrate-producing microbes in humans has been associated with an increased risk of metabolic disease, showing its strength in mitigation of the metabolic disturbances of obesity [9]. This makes it particularly relevant to consider the microbial origin of this fermentation product and the potential for changes in diet and gut physiology to affect its relative production rates and concentrations. SCFAs concentration in the intestinal lumen varies between 60 and 150 mmol/kg, and their daily production in the large intestine of a healthy individual is 300–400 mmol; assuming a daily production of 9 L of the intestinal content, among SCFAs the physiological concentration range of the C-4 fatty acid butyrate is 9–90 mmol/day (1–10 mmol/L) [19]. This daily demand of butyrate (1–10 g/day) should be covered by the fermentation processes of RS and food fibers, but the insufficient intake of these foodstuffs in Western population could represent the link of the rapid growing of many chronic non transmissible conditions. Butyrate is a functionally extremely versatile molecule potentially useful for the prevention and treatment of several metabolic diseases. Here, we review several mechanisms that contribute to explain the role of butyrate in this context.

## 2. Butyrate

### 2.1. Factors that Promote/Inhibit Butyrate Production

The importance of SCFAs for human health has been demonstrated in many studies. The production of these acids in sufficient quantities by the GM is essential for the health homeostasis and well-being of the host [20]. However, the production of these metabolites depends on the GM’s structure, whose function/composition is highly variable and significantly influenced by various factors, such as age, genetics, lifestyle and environment [21]. Hygiene and the use of antibiotics, together with the “Westernization” of lifestyles are associated with an imbalanced GM, or dysbiosis, which may lead to suppression of butyrate production [22,23,24]. Voluptuous habits like smoking or drinking alcohol could also drive to a reduction of butyrate production, due to a decrease of its fermenting bacteria producers in the GM [25,26]. Different dietary patterns, in particular of macronutrients and micronutrients diet composition and the nutritional sources of foods, contribute to GM remodeling. Among macronutrients, CHO play the most crucial role in shaping the bacterial community and their effects on GM have been the most described; according to their degree of polymerization (DP), they can be divided into sugars (DP 1–2), oligosaccharides (short-chain carbohydrates) (DP 3–9) and polysaccharides (DP > or = 10) and based on their capability to be enzymatically degraded in the small intestine, in digestible and non-digestible CHO [27]. It has been evidenced that simple sugars (e.g., sucrose, fructose) cause rapid GM deregulation and hence metabolic dysfunction in the host [28,29]; moreover contrary to popular belief, non-caloric artificial sweeteners (e.g., saccharin, sucralose, aspartame, cyclamate, neotame and acesulfame potassium), which are promoted as substitutes of foods and drinks rich in sugars calories, may be considered unhealthy for reporting both dysbiosis and disruption of metabolic homeostasis, such as glucose intolerance in a GM-dependent manner [30]. While, non-digestible CHO, specifically fiber and RS are beneficial for GM composition; in particular, fiber is a good source of MACs, necessary substrate for butyrate production as reported above, moreover a recent systematic review and meta-analysis found that dietary fiber beneficially modulates the metabolic outputs of the GM, likely due to cross-feeding interactions between butyrate producers and *Bifidobacterium* and *Lactobacillus* species [31]. Among macronutrients, proteins represent an essential source of daily energy for human diet. High dietary protein intake leads to a disproportionate decrease in faecal butyrate with a markedly decline in butyrate-producing *Roseburia*/ *Eubacterium rectale* group of bacteria populations [32]. Several dietary patterns, including Western, gluten-free, omnivore, vegetarian, vegan, and Mediterranean, have been studied for their ability to modulate the GM; among these a high adherence to the Mediterranean diet (MD) was associated with a beneficial microbiome-related metabolomic profile (high *Prevotella* and certain fiber-degrading *Firmicutes* profiles, high SCFAs production) [33]. The increased intestinal SCFAs level induced by MD diet is determined by high consumption of vegetables, fruits and legumes, which are rich sources of complex and insoluble fiber, the major substrates for microbial production of SCFAs. These foodstuffs are typical of vegan/vegetarian diet, in which SCFAs production is usually increased; farther, typical constituents of plant foods are polyphenols, a broad group of substances (such as catechins, flavonols, flavones, anthocyanins, phenolic acids) with well-described antioxidant properties. Previous studies have shown that polyphenols have a beneficial effect on GM by increasing the abundance of *Bifidobacterium* and *Lactobacillus* and elevating the production of SCFAs [13,34]. Habitual dietary pattern can shape the structure and function of the GM, influencing the availability of the GM metabolite butyrate. In fact, a Western diet quickly modifies the composition and metabolic activity of the GM, as demonstrated in the animal model by Carmody and co-workers. In fact, they demonstrated that the consumption of a high-fat and high-sugar diet altered the GM in a linear dose response way, taking an average time of 3.5 days [35]. Although it has previously been suggested that the human GM under normal conditions is relatively stable, other studies have shown that even in adults the microbiota composition can be highly variable and changes over the course of a day [36] or weeks [37]. Behavioral factors, including timing of eating and overnight-fast duration, were also predictive of bacterial abundances. Kaczmarek and co-workers showed that SCFAs concentrations decrease over the course of the day, in particular butyrate, because the amount of *Roseburia* and *Eubacterium*, bacteria producing butyrate, decreased throughout the day [36]. Previously, it has been demonstrated that butyrate has a rhythmically behaviour in murine models [38]. However, only few human studies have primarily emphasized changes in gut microbial diversity (for example, *Bacteroidetes*-to-*Firmicutes* ratio) following dietary intake of RS or other fermentable CHO [39,40,41]. Some interesting works have underlined that rapid changes in microbial metabolic activity and diversity related to the protein, lipid and dietary fibers content of the diet [42,43]. In particular, David and co-workers have also observed significant positive correlations between clusters, such as *Roseburia*, *E. rectale*, and *F. Prausnitzii* that are butyrate producers, and the results of CHO fermentation [42]. In Figure 1 are represented the factors that promote/inhibit butyrate production through a positive or negative modulation of the GM.

### 2.2. Dietary Sources of Butyrate

Dietary sources by which bacterial fermentation produced butyrate are non-digestible CHO, including dietary fibers, resistant starch (RS) and complexes of amylose and/or amylopectin [44,45,46]. RS naturally occurs in foods such as cooked and cooled potatoes, raw bananas, legumes, plant material and partly milled seeds [47]. RS can also be incorporated into breakfast cereals, tortillas, breads and corn (maize) through manufacturing techniques as well as fortification [47,48]. Farther, albeit to a small extent, butyrate is also formed as fermentation product of peptides and amino acids of protein foods: although the numbers of amino acid-fermenting bacteria have been estimated to constitute less than 1% of the large intestinal microbiota, SCFAs represent their catabolism metabolites [17,49]. As such, butyrate occurs in dairy products in considerable amounts, e.g., butter (∼3 g/100 g), cheese (especially goat’s cheese (∼1–1.8 g/100 g), parmesan (∼1.5 g/100 g), whole cow’s milk (∼0.1 g/100 g), due to microbial anaerobic fermentation of fibers, including cellulose, in the ruminant gut [50]. Human milk (HM) has also been examined as a potential source of butyrate for infants, contributing to their colonic microbiota modulation [51,52,53]. Evidences support the hypothesis that HM butyrate could be or derive from the maternal entero-mammary circle or could be produced by the mammalian gland breast milk microbiome: this hypothesis is supported by recent observations demonstrating the presence of potential butyrate-producer bugs [53,54,55,56,57,58]; moreover, a potential pathway in HM butyrate production could derive by HM oligosaccharides (HMOs) metabolization by selected bacteria in the breast [56,59].

### 2.3. Endogenous Production of Butyrate

There are two pathways for the formation of butyrate from butyryl-CoA, (1) the butyryl-CoA:acetate CoA-transferase (*butCoAT*) pathway, which relies on the presence of exogenous acetate, and (2) the phosphor-transbutyrylase/butyrate kinase (*buk*) pathway [17]. Butyrate production is widely distributed among phylogenetically diverse human colonic Gram-positive *Firmicutes* [60]. Although numerous bacterial strains have been analyzed for their butyrate-producing capacities, *Faecalibacterium prausnitzii* (a member of Clostridium cluster IV) and *Eubacterium rectale/Roseburia* (Clostridium cluster XIVa) have currently received the most attention, as they constitute 5–10% of total bacteria in fecal samples collected from healthy adults [60]. In addition to the colonization of the colon by butyrogenic bacteria, it has been proposed that cross-feeding interactions between *Bifidobacterial* strains and *F. prausnitzii* may ultimately enhance butyrate production [61]. Through the production of butyrate, GM could modulate also neurodevelopmental health/disease. The importance of the gut-brain axis has been widely described and a growing body of evidence highlights a role for alterations in the GM in the pathophysiology of many neurologic and neuropsychiatric diseases, including autism spectrum disorder (ASD). Hua and co-workers reported a significantly reduction in the abundances of butyrate-producing bacteria *Faecalibacterium* and *Agathobacter* in ASD children [62]. To further emphasize this link recent work has shown that GM alterations leads to have a significant impact on amygdala development in infancy, modulating the development of the frontal cortex and other brain regions [63].

### 2.4. Physiological Intestinal, Portal and Systemic Butyrate Concentrations

The 20% of total SCFAs produced in the colon consists of butyrate, whose production is 14,700–24,400 μmol per kg of luminal content in the colon, and it represents only the 8% of the total SCFAs in portal vein (15–30 μmol/L) [64]. Compared to colonic levels, human plasma concentration of butyrate is <10 μM [65,66,67,68,69], because butyrate is the preferred energy source for colonocytes and the majority of absorbed may already be consumed by the colonic mucosa. Bloemen and coworkers showed that the release of butyrate from the liver is not significantly different from zero, suggesting a splanchnic (gut + liver) extraction of ∼100% [70]; more recently, the quantification of the percentages of acetate, propionate and butyrate originating from the colon that reached the systemic circulation was evaluated in healthy subjects through the direct stable isotope approach that quantified butyrate systemic availability (2%) and its splanchnic extraction that was of 98% [71].

### 2.5. Butyrate Absorption

All SCFAs are rapidly and almost completely absorbed by the enterocytes and delivered through the portal vein into the liver and systemic circulation. Likely the other SCFAs, butyrate is readily absorbed through a non-ionic diffusion across the apical mem-brane of colonocytes for its hydrophobicity and low molecular weight [72,73]. However, since 10% of SCFAs are revealed in the feces, there may be another mechanism for their uptake [74]. It has been speculated that sodium-coupled monocarboxylate transporter 1 (SCMT1) is able to sequester SCFAs within colonocytes utilizing the colonic Na^+^ concentration gradient [74,75,76], but the solute carrier family 5 member 8 (SLC5A8) is the primary transporter of butyrate across the apical membrane of the colonocytes. SMCT1 has a preference for butyrate and transports propionate and acetate at a slower rate, although it is probably still the main transporter for them [77,78]. It may also be the crosstalk node between the microbiome and the host [72,76]. It has been proposed that proton-coupled monocarboxylate transportation and SCFA-bicarbonate antiporters could be a viable mechanism for SCFAs uptake as well as regulators of lumen pH [79,80,81,82]. The overlap among SCFAs in absorption is not surprising, since butyrate metabolism satisfies ∼70% of colonocytes energy requirements [71,72,83].

### 2.6. Cellular Signaling Pathways of Butyrate

Butyrate is the ligand for metabolite-sensing G-protein coupled receptors (GPCRs), such as GPR43, GPR41 and GPR109A [84,85,86,87]. GPR43 and GPR41 are known as free fatty acid receptors 2 (FFAR2) and 3 (FFAR3), respectively [85,88]. GRP41/FFAR3 receptors are expressed in peripheral nerves, enteroendocrine L and K cells, white adipocytes, pancreatic β- cells, thymus and myeloid dendritic cells; GRP43/FFAR2 receptors are expressed in white adipocytes, enteroendocrine L cells, intestinal epithelial cells, pancreatic β- cells, and several immune system cells such as colonic T-regulatory (Treg) cells, M2 macrophages, neutrophils, eosinophils and mast cells [89]. In particular, FFAR2 can activate both the Gi/o and the Gq pathways, while GPR109A and FFAR3 can only utilize the Gi/o pathway [88,90]. For the Gq pathway, DAG and IP3 may also activate protein kinase C (PKC) to stimulate the downstream activities of the extracellular signal-regulated kinase ½ (ERK-1/2) and c-Jun N-terminal kinase (JNK) pathways [91]. FFAR2 and FFAR3 activation regulates inflammatory pathways such as that are important in determining gut health, such as mitogen-activated protein kinase (MAPK) signaling [92,93]. Butyrate GPCR activation in the gut induces the secretion of endocrine hormones glucagon-like peptide 1 (GLP-1) and peptide YY (PYY) [87,94,95]. It is largely known that GLP-1 increase insulin secretion and that FFAR2 knockout (KO) mice have decreased serum insulin levels [94], while PYY influences energy intake and expenditure at hypothalamus and brainstem levels and FFAR3 KO mice have very low PYY expression [96,97,98]. In particular, butyrate-induced the up regulation of GLP-1 and PYY may be important in preventing or treating obesity and insulin resistance. On the contrary, GPR109A increase glucose uptake and may contribute to hyperglycemia, obesity and insulin resistance [99,100]; it has also been observed that GPR109A has an age-dependent increased expression in the jejunum of diabetic mice. Butyrate is also a histone deacetylase inhibitor (HDACi), targeting class I and II HDACs [101,102], so it may regulate epithelial cell gene expression through epigenetic mechanism involving chromatin remodeling as well as through targeting and regulation of non-histone proteins [101]. As HDAC inhibitor, butyrate may have a potential in the therapy of these conditions improving synaptic plasticity and cognition deficits. Recently has been demonstrated that butyrate has positive effects on social behavior in an ASD mouse model [103], and in general, butyrate can enhance mitochondrial function in the context of physiological stress and/or mitochondrial dysfunction [104]. In particular, Berding and Donovan have underlined that GM composition of children with ASD could vary over time and that diet might play an important role in determining stability of GM in this population [105]. Because of the pleiotropic intracellular signaling effects of butyrate, the physiologic effects of butyrate are multivariate, with outcomes dependent upon tissue type, dosage and time effects. It has been reported that butyrate is able to influence the expression of colonic tight junction (TJ) proteins including claudin-2, occludin, cingulin, and zonula occludens proteins (ZO-1, ZO-2) [106]. Butyrate also facilitated the association between transcription factors and the claudin-1 promoter [107], increased AMP-activated protein kinase (AMPK) activity [108] and reduced bacterial translocation [109]. It was reported that butyrate could also influence the function of the intestinal barrier activating peroxisome proliferator-activated receptor γ (PPAR-γ) [108,110] and modifying transporters expression in vivo and in vitro [111], including MCT1 and MCT4 [112]. Several studies have reported that PPAR-γ is also involved in colon cell differentiation and colon cancer [113,114]. Byndloss and co-workers have found out that a depletion of butyrate-producing microbes by antibiotic treatment reduced epithelial signaling through the intracellular butyrate sensor PPAR-γ [115]. Nitrate levels increased in the colonic lumen because epithelial expression of Nos2, the gene encoding inducible nitric oxide synthase (iNOS), was elevated in the absence of PPAR-γ-signaling. Microbiota-induced PPAR-γ-signaling also limits the luminal bioavailability of oxygen by driving the energy metabolism of colonocytes towards β-oxidation. Therefore, microbiota-activated PPAR-γ-signaling is a homeostatic pathway that prevents a dysbiotic expansion of potentially pathogenic *Escherichia* and *Salmonella* by reducing the bioavailability of respiratory electron acceptors to *Enterobacteriaceae* in the colonic lumen. Furthermore, butyrate has anti-inflammatory effects that result from inhibition of activation of the transcription factor κB (NF-κB), and consequent reduced formation of proinflammatory cytokines [116,117]. Depending on its concentration, butyrate can inhibit growth or promote differentiation of human cells in tissue culture. It can also induce apoptosis in tumor cells, while also acting as atrophic factor for cells in intact tissues [118]. All these interactions here reported have important consequences for the health of the colonic epithelium and for the host homeostasis. In Figure 2 there is a schematic illustration of the direct and indirect effects of butyrate on the regulation of host metabolic functions.

## 3. Butyrate as a Regulator of Body Weight

### 3.1. Butyrate and Energy Expenditure

Modulating the energy intake-expenditure balance in the body is one of the best strategies for obesity therapy. Butyrate could act as a regulator of body weight: a reasonable speculation is that it acts on components of the energy balance, promoting energy expenditure and/or reducing energy intake. Butyrate supplementation was found to have multiple metabolic benefits, including prevention of high-fat diet (HFD)-induced obesity and obesity-associated disorders in the animal model [119,120,121,122,123,124]. In addition to its preventive actions in diet-induced disease models, butyrate is also effective in treating obesity through the promotion of energy expenditure and the induction of mitochondrial function: the mechanism of butyrate action for obesity is related to the activation of AMPK, increasing ATP consumption, and the induction of PGC-1α activity, the molecular mechanism by which butyrate stimulates mitochondrial function in association with up-regulated expression of genes involved in lipolysis and fatty acid β-oxidation [125,126,127]. Adipose tissue is an endocrine organ which constitutes the largest energy reservoir in the body and plays an important role in energy homeostasis. Thus, increasing fat mobilization in adipose tissue is an attractive potential strategy for the management and treatment of obesity. Butyrate-mediated regulation of thermogenesis and energy homeostasis was also demonstrated in a recent study performed in mice: it promotes thermogenesis in brown adipose tissue (BAT) through the activation of lysine specific demethylase (LSD1), a histone demethylase, important regulator of thermogenesis [128]. Farther, β-adrenergic receptors that are largely expressed in adipose tissue, play a fundamental role in lipolysis. Another speculation by which butyrate could induce fat burning is β3-adrenergic receptors activation in mice white adipose tissue, which may encourage potential anti-obesity application of butyrate in humans [129].

### 3.2. Butyrate and Energy Intake

Evidence suggest that butyrate inhibit weight gain via suppressing food intake: Li and coworkers showed that oral administration of butyrate in mice induces satiety and reduces cumulative food intake, suppressing the activity of orexigenic neurons that express neuropeptide Y in the hypothalamus indicating a mechanism involving the gut-brain neural circuit [130]. In addition, the hypophagic effect of butyrate can be explained through the increase of anorexigenic gut hormones that directly act on the hypothalamus to regulate satiety signaling; in fact, several animal studies showed that oral butyrate has the capacity to evoke an anorectic response increasing plasma levels of GLP-1, glucose-dependent insulinotropic polypeptide (GIP) and PYY [94,131]. In a randomized clinical trial, the colonic infusion of butyrate as SCFAs mixtures increased fat oxidation and fasting and postprandial plasma PYY concentrations in overweight/obese humans [132]. In the context of human studies, another interesting one conducted in a group of obese volunteers with metabolic syndrome showed that the oral supplementation with 4 g of sodium butyrate per day decreased oxLDL-induced trained immunity for LPS-induced IL-6 responses and Pam3CSK4-induced TNF-α response, showing a positive anti-inflammatory and immunomodulatory effect and possibly slowing down the process of vascular wall inflammation and the progression of atherosclerosis, one of the metabolic complications caused by obesity [133]. Butyrate also acts through the regulation of the opioidergic system and therefore with subjective pleasure and dysphoria that guide food intake. Especially via reward regulation, alterations in the opioidergic system are intimately associated with food intake dysregulation in obesity. Decrease in µ-opioid receptor levels have been proposed to drive food intake in obese individuals, but butyrate epigenetically upregulates the µ-opioid receptor whose activation is classically associated with reward [134]. Based on the available data butyrate can be seen as a novel strategy to improve long-term energy homeostasis.

## 4. Butyrate and Liver

Obesity is closely related with liver metabolism disorders, such as non-alcoholic fatty liver disease (NAFLD), a hepatic lipid-related metabolic disorder that represents a spectrum of liver derangements starting from benign steatosis that progresses to non-alcoholic steatohepatitis (NASH) characterized by insulin resistance, liver inflammation and fibro-sis [135].

### 4.1. Butyrate and Lipids Metabolism Impairment

An impairment of lipidemia appears to have a central role in the pathogenesis and progression of the metabolic disorders associated with obesity. Butyrate could regulate lipid metabolism in the liver and intestine and several findings showed that it exerts beneficial effects on liver diseases. It is able to downregulate the expression of nine key genes involved in the intestinal cholesterol biosynthesis pathway and thereby it may inhibit hypercholesterolemia [136,137]. Cholesterol homeostasis is a dynamic balance between biosynthesis, uptake, export and esterification: when cholesterol is in surplus it is esterified and stored as reservoir in cytosolic lipid droplets or released as a major constituent of plasma lipoproteins, including chylomicrons, VLDLs, LDLs and HDLs [138]. Butyrate modulates the cellular events governing the assembly and delivery of lipoproteins as demonstrated in a decrease secretion of chylomicrons and VLDLs in butyrate-treated Caco-2 cells [139]. The reverse cholesterol transport (RCT) has a role in atherosclerosis development, promoting the movement of cholesterol from the peripheral tissues to the liver for reuse or for the final elimination into the bile and retarding the plaque progression. Butyrate may accelerate RCT and thus preventing atherosclerosis development, by activating ATP-binding cassette sub-family A member 1 (ABCA1) expression, that promotes cholesterol efflux to lipid-free/poor apolipoprotein A–I, both in liver and in plaque (peripheral macrophages), via a specificity protein 1 (Sp1) pathway, as shown in HFD-induced Apolipoprotein E deficiency (ApoE-/-) mice model [140]. Scavenger receptor class B type I (SR-BI), the first molecularly well-defined HDL receptor in mice, and its human homologue CLA-1 (CD36 and Lysosomal integral membrane protein-II Analogous-1) plays a pivotal role both in the initial (cholesterol efflux and removal from the artery wall) and final (selective HDL-cholesterol uptake in the liver) phase of RCT; it has previously shown that butyrate might enhance the transcriptional activity of SR-BI/CLA-1 expression in human hepatoma HepG2 cells, facilitating RCT with an antiatherogenic activity [141].

### 4.2. Butyrate and Liver Disorders

Butyrate administration ameliorates hepatic steatosis induced by HFD in mice, through a reduction in intrahepatic lipid accumulation (triglycerides and phospholipids content) and reduction of liver weight [118,129,142]. In-depth mechanistic investigation focused on the liver showed that hepatic mitochondria are the main target of the beneficial effect of butyrate in reverting fat accumulation in diet-induced obesity [143]. Hepatic mitochondrial dysfunction is a key pathway for the alteration of fat oxidation, reactive oxy-gen species production, oxidative stress, and the progression of NAFLD [144]. Sun and coworkers showed that butyrate up-regulated hepatic expression of peroxisome proliferator-activated receptor α (PPARα), essential regulator for mitochondrial fatty acid oxidation, alleviating HFD-induced NAFLD in rats, through the activation of β-oxidation and inflammation suppression [145]. Butyrate is able to attenuate inflammatory mediators re-leasing in liver [146]; this effect indicated that butyrate promotes the maintenance of liver homeostasis, with the decrease of many pro-inflammatory factors (MCP-1, TNF-α, IL-1, IL-2, IL-6, IFN-γ) and the increase of anti-inflammatory ones (IL-4, IL-10), that could occur through the immunoregulation of via inhibiting histone acetylation enzymes or GPCRs pathway, as shown in HFD-induced steatohepatitis mice model [147]. Through the activation of GPCR (GLP-1R), GLP-1 exerts its effects in lipid metabolism, protecting hepatocytes against steatosis [148,149]: HFD and lipotoxicity, typical of obesity, induce the loss of GLP-1 responsiveness, but butyrate has the capability to upregulate hepatic GLP-1R expression, with a decrease in fatty acid synthesis lead by the involvement of (AMP)-activated protein kinase/AcetylCoA carboxylase (AMPK/ACC) signaling [150,151].

## 5. Butyrate and Type 2 Diabetes

The worldwide occurrence of diabetes (both type-1 and type-2) is predicted to rise to 629 million by 2045 [152]. Obesity is a predisposing element in the development of type 2 diabetes mellitus (T2DM) which accounts for 90–95% of all diabetes cases [153]. The primary induction driver of numerous life-threatening complications and comorbidities is the dysregulation of glucose metabolism (fasting and postprandial hyperglycemia), with many deleterious downstream effects such as chronic inflammation, cardiovascular dis-eases, gut microbial dysbiosis, with an enhancement in opportunistic pathogens and a reduction of many butyrate-producing bacteria [154]. However, this glycemic alteration can be followed to progressive impairments in insulin sensitivity (i.e., insulin resistance) and a corresponding failure of pancreatic islets to maintain appropriate insulin output to compensate for the decline in insulin sensitivity (i.e., islet failure) [155]. Islets are known to express the butyrate receptors GPR41 and 43, indicating that it might be involved in islet-cell metabolism and function [156,157,158,159], as demonstrated by the effects of butyrate pre-incubation in improving diabetes-induced histological alteration of islet and functional damage [160,161]. There has been revealed an anti-diabetogenic effect of butyrate in animal models of T2DM, which was related to its action as HDAC inhibitor [162]. HDAC inhibition ameliorates hyperglycemia by controlling the expression of glucose-6-phosphate and the subsequent gluconeogenesis [163].

### Butyrate and GLP-1 Secretion

Butyrate is able to stimulate the release of GLP-1 from intestinal L-cells as shown both in cell culture system and animal model [164,165]. GLP-1 has the ability to reduce apoptosis and to induce pancreatic β-cells neogenesis and regeneration, as well as to induce insulin synthesis and secretion; GLP-1 activates the GLP-1R GPCR in β cells, resulting in production of cAMP, which in turn leads to an increase in glucose-stimulated insulin secretion [166]. Preclinical data are also confirmed by a randomized, double blind, clinical placebo control trial on type 2 diabetic adults, where was shown a significant increase in postprandial GLP-1 concentration after butyrate supplementation and a decreasing trend was observed for homeostatic model assessment of insulin resistance (HOMA-IR) indices [167]. Interestingly, oral butyrate treatment beneficially affects glucose metabolism in lean subjects, who benefited from an improvement in both peripheral and hepatic insulin sensitivity at a dosage intake of 4 g of sodium butyrate daily for 4 weeks [65]. At this juncture, an interesting speculation may be to develop butyrate-based therapies for diabetic or pre-diabetic patients to exploit its beneficial effects in the context of obesity-related glucose metabolism impairment.

## 6. Conclusions

Butyrate is a functionally extremely versatile molecule produced by human GM. Host metabolism and immune function are critically regulated by butyrate. This implicates butyrate as a key mediator of host-microbe crosstalk, as well for its ability to regulate different metabolic pathways at the same time. The Table 1 summarizes main preclinical and clinical data on the butyrate’s effects in the context of obesity and related metabolic diseases.

Increasing endogenous butyrate production could be a valuable strategy in the prevention of obesity and related metabolic diseases, but also increasing exogenous intake through butyrate supplements. Most likely, the causative lack of randomized controlled trials proving the efficacy of butyrate in these metabolic disorders is mainly due to the poor palatability of the actual butyrate preparations available on the market. Nevertheless, there is an urgent need for products that mask the unpleasant organoleptic properties of butyrate, in order to facilitate clinical studies in children and in adult patients.

## Figures and Tables

**Figure 1 molecules-26-00682-f001:**
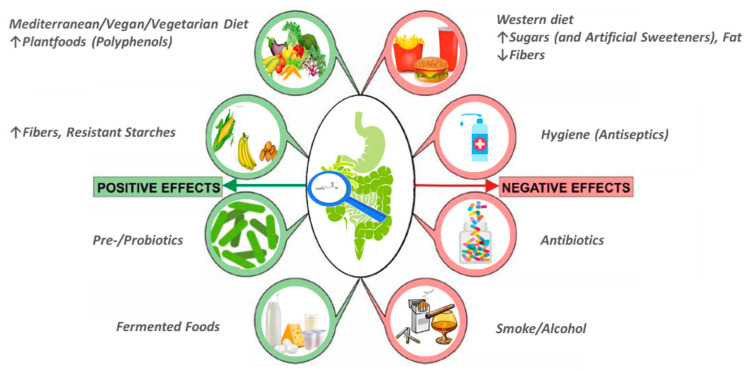
Factors that promote/inhibit butyrate production through a positive or negative modulation of the gut microbiota (GM). Among negative factors, hygiene and the use of antibiotics, together with the “Westernization” of lifestyles (high consumption of fats and sugars, sedentary lifestyle) are associated with an imbalanced intestinal microbiota, or dysbiosis, which may lead to suppression of GM fermentation and butyrate production. Farther, voluptuous habits like smoking or drinking alcohol could also drive in a reduction of butyrate production, due to a decrease of its fermenting bacteria producers in the GM. Within the factors that positively affect the production of butyrate there are dietary patterns characterized by a high consumption of plant foods (Mediterranean, vegan/vegetarian diet), source of fiber, resistant starch and polyphenols, necessary substrates for butyrate production. Moreover, among positive modulator arise nutritional supplementation with prebiotics (that can serve as the substrates for bacterial fermentation in the colon to generate butyrate) and/or probiotics, live cultures of specific strains of bacteria that colonize the intestinal tract to promote generation of butyrate (food sources of probiotics are yogurt, fermented cheeses etc.).

**Figure 2 molecules-26-00682-f002:**
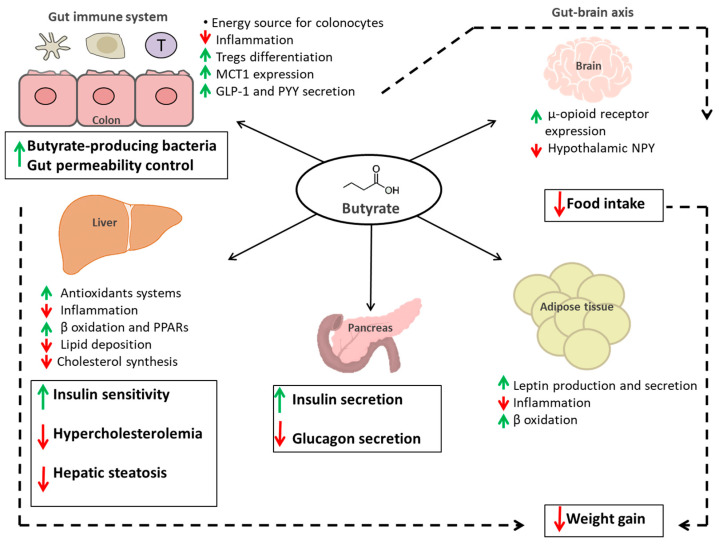
Schematic illustration of the direct (continuous black arrow) and indirect (dotted black arrow) effects of butyrate on the regulation of host metabolic functions. Butyrate positive effects are indicated with a green arrow, negative ones are indicated with red arrows. Abbreviations: Tregs: T regulatory cells; MCT1: Monocarboxylate transporter 1; GLP-1: Glucagon-like peptide 1; PYY: Peptide YY; NPY: Neuropeptide Y; PPARs: Peroxisome proliferator-activated receptors.

**Table 1 molecules-26-00682-t001:** Main preclinical and clinical evidences on the protective role of butyrate against obesity and obesity-related diseases.

Effect	Preclinical	Clinical
↑ fat oxidation; ↑ fasting and postprandial plasma PYY concentrations		[118]
Prevention of HFD-induced obesity, improvements in obesity-related lipid accumulation and low-grade chronic inflammation (↓serum LPS concentrations)	[106]	
Recovering HFD-induced changes in body weight, adiposity, liver pathology	[107,108]	
Prevention of HFD-induced increase in body weight and adiposity; liver inflammation and damage, steatosis, impairment of glucose homeostasis and the onset of IR	[108,109]	
Promotion of energy expenditure and mitochondrial function induction (AMPK activation; induction of PGC-1α activity; ↑expression of genes involved in lipolysis and fatty acid β-oxidation; alleviating diet-induced obesity through activation of ARβ3-mediated lipolysis in WAT	[113,115]	
Shifting metabolism in adipose and liver tissue from lipogenesis to fatty acid oxidation (downregulation of PPARγ activating an UCP2-AMPK/ACC network)	[112]	
Alleviating HFD-induced obesity (through activation of adiponectin-mediated pathway and stimulation of mitochondrial function in the skeletal muscle)	[111]	
Induction of thermogenesis in BAT and scWAT (LSD1 activation)	[114]	
Action on the gut-brain neural circuit to improve energy metabolism via ↓energy intake and ↑fat oxidation by activating BAT	[116]	
Gut hormone release regulation (↑GLP-1, GIP), ↓food intake, diet-induced obesity protection	[83,117,150,151]	
↓oxLDL-induced trained immunity for LPS-induced IL-6 responses and Pam3CSK4-induced TNF-α responses		[119]
↓expression of nine key genes involved in the intestinal cholesterol biosynthesis pathway; hypercholesterolemia inhibition	[122,123]	
↓secretion of chylomicrons and VLDLs	[125]	
Ameliorating HFD-induced atherosclerosis (via ABCA1-mediated cholesterol efflux in macrophages)	[126]	
Facilitating reverse cholesterol transport with an antiatherogenic activity (↑SR-BI/CLA-1 expression)	[127]	
Suppression of HFD-induced liver weight gain and hepatic TGs accumulation; improvement of hepatic metabolic conditions via FFAR3	[128]	
Ameliorating HFD-induced NAFLD through PPARα- mediated activation of fatty acid β oxidation and inflammation suppression (↓NF-κB)	[131]	
Promotion of liver homeostasis (↓pro-inflammatory and ↑anti-inflammatory factors)	[132,133]	
↑fatty acid oxidation, ↓fatty acid synthesis (via AMPK/ACC pathway)	[129,137]	
Anti-diabetogenic effect (↑GLP-1 secretion; ↑insulin sensitivity;	[148,150,151]	[54,153]
↑phosphorylation of insulin receptor; ↑GLUT2 expression)	[129]	
↑diabetes-induced histological alteration of pancreatic islet and functional damage	[146,147]	

PYY: peptide YY; HFD: high fat diet; LPS: lipopolysaccharides; IR: insulin resistance; AMPK: adenosine 5′-monophosphate-activated protein kinase; PGC-1α: peroxisome proliferator–activated receptor (PPAR)-γ coactivator (PGC)-1α; ARβ3: β3-adrenergic receptor; WAT: white adipose tissue; PPARγ: peroxisome proliferator–activated receptor- γ; UCP2: mitochondrial uncoupling protein 2; AMPK/ACC: AMP-activated protein kinases/acetyl coenzyme A carboxylase; BAT: brown adipose tissue; scWAT: subcutaneous white adipose tissue; LSD1: lysine specific demethylase; GLP-1: glucagon-like peptide 1; GIP: glucose-dependent insulinotropic peptide; oxLDL: oxidized low-density lipoprotein; IL-6: interleukin 6; TNF-α: tumor necrosis factor α; VLDLs: very low-density lipoproteins; ABCA1: ATP-binding cassette sub-family A member 1; SR-BI: scavenger receptor class B type I; CLA-1: CD36 and lysosomal integral membrane protein-II analogous-1; TGs: triglycerides; FFAR3: free fatty acids receptors 3; NAFLD: non-alcoholic fatty liver disease; PPARα: peroxisome proliferator-activated receptor α; NF-κB: nuclear factor κB; GLUT2: glucose transporter 2.

## Data Availability

No new data were created or analyzed in this study. Data sharing is not applicable to this article.

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
