# Peer review of "The Protective Role of Butyrate against Obesity and Obesity-Related Diseases"

_molecules, 2021, doi:10.3390/molecules26030682_

Round 1
Reviewer 1 Report
This is an interesting paper. the authors should add some recent review about the role of butyrate in autism spectrum disorder. how much an impaired, not regulardietary habit can affect endogenous butyrate production or its activity, due to change in gut microbiome composition?
Reviewer 2 Report
Interesting work!
The Authors report all the effects of a short-chain fatty acid, Butyrate, primary end product of degradation of dietary fibers by gut microbiota. Contextually, the Authors, explain the possible protective role of Butyrate against obesity and obesity-related diseases by providing the main preclinical and clinical evidences. The Authors seem to have a great experience in this field.
The manuscript is very well articulated and structured in its sections; design, search strategy, keywords and reported data seem adequately corrected and suitable to scientific rigour; the scientific contents are supported by valid References and the narrative is comprehensive and adequately clear/understandable to common Reader; figures and table are also clear.
I think this work globally is very interesting and useful for further developments and progress in management, prevention and treatment of weight gain and its complications.
I would suggest the Authors to replace the term “Gut Microbioma” with the term “Gut Microbiota” on page 2, lines 60-61; I think “microbiota” is more appropriate. I have not other suggestions to give to the Author nor other specific comments, except to look for potential typing mistakes to be corrected.
Whereupon the manuscript can be accepted for publication; this is my opinion.c
Reviewer 3 Report
In this paper, the authors summarize recent studies on butyrate, regarding its sources, and physiological and pathophysiological significances. I think this review is worthwhile to the researchers studying butyrate.
Minor comment:
- In page 1, line 29, is “reviewedthe” ”reviewed the”?
- In page 11, line 467, is “Interleukin” ”interleukin”?
- In page 11, line 469, is “Lysosomal integral membrane protein-II Analogous-1” ”lysosomal integral membrane protein-II analogous-1”?
